# Factors and Processes Facilitating Recovery from Coercion in Mental Health Services—A Meta-Ethnography

**DOI:** 10.3390/healthcare12060628

**Published:** 2024-03-11

**Authors:** Lene Lauge Berring, Eugenie Georgaca, Sophie Hirsch, Hülya Bilgin, Burcu Kömürcü Akik, Merve Aydin, Evi Verbeke, Gian Maria Galeazzi, Stijn Vanheule, Davide Bertani

**Affiliations:** 1Psychiatric Research Unit, Psychiatry Region Zealand, 4200 Slagelse, Denmark; 2Department of Regional Health Research, University of Southern Denmark, 5230 Odense, Denmark; 3School of Psychology, Aristotle University of Thessaloniki, 54124 Thessaloniki, Greece; georgaca@psy.auth.gr; 4Department for Psychiatry and Psychotherapy I, Faculty of Medicine, Ulm University, 89081 Ulm, Germany; sophie.hirsch@zfp-zentrum.de; 5Department for Psychiatry and Psychotherapy Biberach, ZfP Südwürttemberg, 70597 Stuttgart, Germany; 6Mental Health and Psychiatric Nursing Florence Nightingale Nursing Faculty, Istanbul University-Cerrahpasa, 34000 Istanbul, Turkey; huliaa.bilgin@iuc.edu.tr; 7Department of Psychology, Faculty of Languages and History-Geography, Ankara University, 06100 Ankara, Turkey; komurcu@ankara.edu.tr; 8Mental Health and Psychiatric Nursing Department, Karadeniz Technical University, 61080 Trabzon, Turkey; merveaydin@ktu.edu.tr; 9Department of Psychoanalysis and Clinical Consulting, University of Ghent, 9000 Gent, Belgium; evi.verbeke@ugent.be (E.V.); stijn.vanheule@ugent.be (S.V.); 10Department of Biomedical, Metabolic, and Neural Science, University of Modena and Reggio Emilia, 41124 Modena, Italydbertani@gmail.com (D.B.); 11Department of Mental Health and Drug Abuse, Azienda USL IRCCS di Reggio Emilia, 42122 Reggio Emilia, Italy

**Keywords:** recovery, coercion, legal rights, debriefing, meta-ethnography, mental health

## Abstract

Background: Being subjected to or witnessing coercive measures in mental health services can have a negative impact on service users, carers and professionals, as they most often are experienced as dehumanising and traumatic. Coercion should be avoided, but when it does happen, it is important to understand how the experience can be processed so that its consequences are managed. Method: A systematic review and meta-ethnography was used to synthesise findings from qualitative studies that examined service users’, staff’s and relatives’ experiences of recovery from being exposed to coercive measures in mental health care settings. We identified, extracted and synthesised, across 23 studies, the processes and factors that were interpreted as significant to process the experience. Results: Recovery from coercion is dependent on a complex set of conditions that support a sense of dignity and respect, a feeling of safety and empowerment. Being in a facilitating environment, receiving appropriate information and having consistent reciprocal communication with staff are the means through which these conditions can be achieved. People employ strategies to achieve recovery, both during and after coercion, to minimise its impact and process the experience. Conclusions: The findings point to the importance of mental health care settings offering recovery-oriented environments and mental health professionals employing recovery-oriented practices, that would empower service users to develop strategies for managing their mental distress as well as their experiences in mental health care in a way that minimises traumatisation and fosters recovery.

## 1. Introduction

Despite its frequent and broad usage in mental health care, coercion remains a controversial subject [1,2]. It constitutes a violation of persons with mental health conditions [3], as it contradicts human rights and ethical principles, such as respect for autonomy, beneficence, nonmaleficence and justice [4]. Coercion refers to any action that forces a person with a mental health condition to behave contrary to their wishes and consists of legal or illegal restriction of people when they are considered to pose a risk to themselves or others [2,5]. Commonly implemented types of formal coercion include involuntary hospitalisation, observation, seclusion and physical, chemical and mechanical restraint [2]. Also, various informal coercive practices, such as restrictions in contacting others and receiving visitors, as well as using persuasion, leverage and threats to regulate service users’ behaviours [6,7,8], seem to be habitually used in mental health care [9].

Globally, there seems to be considerable variation in the use of coercive measures [10,11], both between and within countries [12]. This is compounded by the fact that the use of coercive measures is reported differently and inconsistently, depending on cultures and legislations. For example, in a study of 10 European countries the use of coercive measures was found to range from 21% to 59% [13], while in Pacific Rim countries a variation of 3000-fold was documented in the use of mechanical restraint [14]. Despite national efforts in some countries to reduce coercive measures, it seems that a sustainable reduction is still difficult to achieve [15].

Reportedly, some persons subjected to coercion can be positively affected by it, because they might associate it with hospital care [2,9,16,17]. This may be reinforced when persons with mental health problems are encouraged to understand aggressive and violent behaviours as part of their own mental health conditions [18]. However, feeling coerced is rated as traumatic and negative by most service users and relatives [19,20]. Being subjected to coercion can operate as a trigger of past traumatic memories [21]. A meta-analysis showed that the likelihood of post-traumatic stress disorder (PTSD) in patients following coercive interventions ranged from 25% to 47% [22]. A study of the impact of mechanical restraint on individuals with schizophrenia [23] found that PTSD was more likely to occur if the coercive episode was interpreted as central to their identity. There is evidence that coercion is perceived by those subjected to it as punitive and anti-therapeutic [1,24]. A sense of humiliation [25] and abandonment, as well as feelings of loneliness and helplessness, frequently follow experiences of being subjected to coercion [26]. Individuals may become more aggressive and agitated and exhibit unfavourable post-discharge behaviour [24], and some patients may stop seeking treatment and assistance out of fear of encountering forceful methods [27,28].

Coercion must only be used to keep persons safe and should never be employed as a form of punishment or for convenience [29]. Unfortunately, this is often not the case in everyday mental health care [30]. Even when appropriately used, however, the use of coercion raises important human rights issues [31] and poses ethical dilemmas to mental health professionals, who are caught between the duty to care and the requirement to control persons in their care, when they pose a risk to themselves or others [32]. Apart from the negative physical and psychological effects coercion has on persons subjected to it, the use of coercion negatively impacts those who implement or witness the incident [33,34,35], e.g., staff, co-patients and relatives. Like service users, health care professionals generally dislike the use of coercion and experience distress, sadness, stress, guilt and restlessness after applying these measures [36,37,38]. Even those mental health professionals who are convinced that coercion is beneficial for service users [4] and therefore see it as an integral part of their job declare that they dislike its use [39]. Despite their reported dislike of coercion, professionals show limited engagement in lowering its use [40].

Research on coercion in mental health care has mainly focused on its implementation and its impact, overwhelmingly negative, on those subjected to it or involved in it. Several initiatives to prevent the use of coercive measures are documented in the relevant literature [41,42,43]. Also, debriefing interventions [44,45,46,47] have been developed but have not been fully implemented. Apart from the limited research on these interventions, little is known about how people involved in coercive incidents cope with this experience in order to overcome and recover from it and how mental health care professionals may mitigate its negative impact.

Knowledge of the processes and factors that facilitate recovery from coercion may be beneficial for both care receivers and caregivers; understanding what might mitigate the harmful aspects of coercion can encourage helpful practices during the use of coercion as well as post hoc interventions. Coercion occurs in a social context and a social understanding of recovery is adopted in this study; recovery is understood as a deeply social, unique and shared process in which living conditions, material surroundings, social relations and sense of self evolve in a way which is interdependent with significant others [48].

The objective of the study was, thus, to systematically review and synthesise empirical qualitative studies of persons with mental health conditions, health care providers and relatives’ subjective experiences of recovery from coercion in order to identify what helps them cope with the experience and manage its traumatic impact during and after the event; in other words, we aimed to identify the factors and processes that facilitate recovery from coercion.

## 2. Materials and Methods

This review was designed as a meta-ethnography, developed by Noblit and Hare [49] in the 1980s as a guide to synthesising contradictory concepts. This interpretive method aims to synthesise information obtained from multiple qualitative studies through an analytical approach, rather than a descriptive one, condensing different findings and meanings into broader and comprehensive concepts or ideas that represent a derivative, higher-order set of data [50,51]. Meta-ethnography, applied to medical research, allows new insights into the experiences of service users and health professionals and generates new evidence and perspectives on care [52], as this study aims to do.

The approach builds on seven progressive steps: 1: getting started, 2: deciding what is relevant, 3: reading the studies, 4: determining how the studies are related, 5: translating the studies into one another, 6: synthesising the translations and 7: expressing the synthesis [49]. Throughout the process, but especially for the analytical stages 3–7, we followed the recommendations by [53] that clarify and expand the Noblit and Hare (1988) [49] framework. The review was written in accordance with the Meta-ethnography Reporting Guidance (eMERGe) [54].

### 2.1. Search Criteria and Strategy

#### 2.1.1. Step 1—Getting Started

As our aim was to identify the factors and processes that enable persons involved with or subjected to coercion deal with it in a way that allows them to move forward in their recovery, we decided to include studies that presented first-person perspectives of persons directly involved in coercive practices. We included all categories of persons involved, namely service users who were subjected to or witnessed coercion, their carers as well as mental health professionals who carry out or witness coercive incidents. As we wanted to capture the experiential aspects of coping with coercion, we included only studies that examine experiences regarding coercion, not views and opinions about it. We decided to include only qualitative studies, as they are better at capturing subjective experience in depth and in an open-ended, exploratory way.

#### 2.1.2. Step 2—Deciding What Is Relevant

The search strategy was led by LLB and developed through authors’ consensus. In accordance with the aim of the review, the search terms included variations of terms related to (a) coercion, (b) mental health/psychiatry, (c) recovery and (d) qualitative methodology. A systematic search on the main scientific databases (Web of Science, Scopus, PsychArticles, PubMed, CiNAHL and Embase) was conducted in June 2021 by a specialist librarian. Table 1 contains the final search string used for the PubMed search, as an example.

Only empirical studies about formal coercion were included, excluding papers about informal coercion, minors, elders, forensic settings, persons with addictions and mentally impaired persons. Qualitative studies, mixed method studies and research case studies were included; surveys, quantitative studies, opinion papers and clinical report studies were excluded.

For screening, 4854 references were uploaded in covidence [55]; 117 of them were removed as duplicates. The first screening by title and abstract was carried out by three pairs of reviewers, with a third reviewer mediating in cases of discordance. After the first screening, 4655 studies were excluded, leaving 80 papers for full-text screening. As we were aware of relevant studies that were not included in the screening database, we decided to conduct an extensive supplementary reference list search. A subgroup of three reviewers (EG, DB and LLB) carefully went through the reference lists of 17 review papers on coercion-related topics and this led to the identification of 91 additional studies. Thereafter, a full-text screening was carried out of these 80 + 91 papers, from which 31 + 16 papers were selected as fulfilling the search criteria.

At this point in the reviewing process, we became fully aware of the novelty of our pursuit. We found no study that focused on or directly addressed our topic. The vast majority of studies examined experiences or opinions of coercion and the negative impact of coercion on those subjected to it, with only passing references to factors which participants considered helpful in coping with the experience of being coerced. This justified our choice of meta-ethnography as the most appropriate method for generating new knowledge in emerging topics by bringing together the limited research on them. Given that meta-ethnography is performed on a small number of meaning-rich papers, we proceeded to further screening through purposive sampling [56]. The author group devised a paper evaluation strategy, summarising what each paper had to say about the factors and processes that enable people to deal with the impact of coercion, and in pairs we ranked these 47 papers in terms of relevance to our pursuit. After broader author group deliberations on the basis of these rankings, we ended up including 23 studies, which provided categories with conceptual depth to allow translation [51,57], for further processing.

The overall process of screening and selection is presented following PRISMA guidelines in Figure 1.

We chose to perform quality appraisal of the studies through the Critical Appraisal Skills Programme [58]; CASP has been successfully adopted in a wide range of meta-ethnographic studies [59,60]. All of the included papers were reviewed individually by BKA and LLB, using the 10-item CASP qualitative checklist to assess their reliability and validity. In case of conflict, consensual discussions with the author team were used for conflict resolution. Detailed evaluation of each CASP domain for all included studies can be seen in the Appendix A.

Main information about the 23 selected papers was extracted into a table (Table 2), including authors, date, country, type of coercion, aim/focus, setting, participants, data collection method and analysis method.

#### 2.1.3. Step 3—Reading the Studies

Most authors working in pairs extracted the meanings that were relevant to the research question, i.e., factors and processes that facilitate recovery from coercion, from each paper in a customised data extraction form. In particular, we identified all the relevant first-order constructs, which were the primary data, from extracts from participants in the selected studies. In a separate column, we recorded second-order constructs, which were the interpretations by the study authors of the participants’ words. On the basis of these two, we devised in a third column of the key concept/theme, a third-order construct that was our interpretation of the meanings provided by the study participants and the study authors [53].

#### 2.1.4. Step 4—Determining How the Studies Are Related

LLB and EG, following a collaborative iterative process, organised the key concepts/themes from all the studies into groups of related themes. They reviewed each group of themes for consistency and examined how the main themes were related within each group, comparing the concepts found in each study with each other, e.g., being in a facilitating environment. They then summarised the meaning of each theme group and devised a representative descriptive title/label, transforming the study themes into meta-ethnographic categories.

#### 2.1.5. Step 5—Translating the Studies into One Another

In the beginning, EG created a table with the frequency of categories found in each paper (Table 3). This allowed us to determine the richest papers in terms of our study question. We ranked the papers accordingly and started the translation and synthesis of each category from the richest papers, gradually incorporating those that had the least to offer to our quest. LLB and EG translated the themes from the studies into one another, within each category, producing both narrative accounts and diagrammatic depictions of the relations found.

#### 2.1.6. Step 6—Synthesising the Translations

Through constant comparison and a reciprocal synthesis, both within each category and between categories, we moved to a line of argument synthesis [53]. At crucial stages of this process, LLB and EG called all-author group meetings and circulated the draft documents to the other authors, with several layers of memos building up and enriching the analysis and synthesis of the meanings. To add rigour to the process and enhance interpretation, the analysis and preliminary findings were presented and discussed with key stakeholders at various scientific and educational fora of the Fostering and Strengthening Approaches to Reducing Coercion in European Mental Health Services (FOSTREN) network that was the context of the current study. We expressed the synthesis both narratively and through the central diagrammatic depictions of the study (see Figure 2).

#### 2.1.7. Step 7—Expressing the Synthesis

The synthesis is expressed in the findings section below.

## 3. Findings

### 3.1. Study Characteristics

Of the 23 studies, 9 examined involuntary hospitalisation, 8 restraint and/or seclusion, 4 community treatment orders (CTOs), 1 locked doors and 1 compulsory medication. The studies were conducted in 10 countries, in Europe, Australia, Canada, China and the US. Participants in 15 studies were service users, staff members solely in 1 study, and a mix of carers, service users and staff in 7 studies. Data collection methods were predominantly interviews, except Ling et al. (2015) [75] and Fletcher et al. (2019) [68], who employed an audit method and a forum, respectively (Table 3).

### 3.2. Synthesising Translations

Recovery from coercion, regardless of the type of formal coercive measures, seems to be dependent on three interwoven and complex sets of factors. The *conditions for recovery from coercion* are experiencing dignity and respect, feeling safe and being empowered. The *means to achieve these conditions* are being in a facilitating environment, receiving appropriate information and consistent reciprocal communication. When these conditions are fulfilled, persons subjected to coercion employ *strategies for dealing with the impact of coercion*, both during coercion, to minimise coercion and its impact, and after coercion, to process the experience. The synthesis is schematically presented in Figure 2.

### 3.3. Conditions for Recovery

It seems that recovery from being subjected to coercion cannot even begin to take place if certain conditions are not present. Below we present what we found to be the conditions for recovery, grouped in three central categories: experiencing dignity and respect, feeling safe and being empowered.

### 3.4. Experiencing Dignity and Respect

Experiencing dignity and respect was an important component, which we extracted from 12 studies (Table 3). It was interpreted in the papers as being met and acknowledged as a person in distress (Figure 3).

Being met and acknowledged as a person in distress means being treated not as a problem but as a patient receiving professional care [72,73,76,83]. This takes place when experiences of distress are taken seriously, listened to and taken care of in a professional manner, and when disruptive reactions leading to coercion are attributed to the illness, not to personal characteristics [78,81,83]. Dignity and respect are linked to being treated not only as a patient but as a person, who is talked to by name. This presupposes that experiences of distress are addressed as distressing experiences, not as meaningless symptoms [65,73], and that staff go beyond these distressing experiences and address common human experiences [61,78,81]. Above all else, it means being treated as human, like any other person, receiving humane care [73,76]. As explained by a study participant who recalled an involuntary admission:
“*I was treated just like any other person that would walk in off the street.*”[26] (p. 1133)

This is achieved when persons subjected to coercion are talked to by name [66] and are engaged with, listened to and taken seriously [26,65,78]. Ultimately, it is linked to the feeling that others have an interest in your thoughts and experiences and converse with you in an ordinary way [61,78,81].

Promoting this feeling of dignity and respect requires staff to be professional and do their job properly but also to go beyond that and engage with service users as people, listening to their thoughts and attending to their needs [73,76]. Humane care entails maintaining meaningful relationships and connections between staff and service users, even during coercive events [73,81]. This happens when physicians not only prescribe medications but also listen to patients and try to understand their feelings, views and concerns [78]. Meaningful relationships and connections occur when staff are friendly and helpful, listening, engaging and taking service users’ views seriously [61,65,75,76,81]. When this happens, service users feel that they receive both professional and humane treatment. Correspondingly, they perceive staff as both competent and caring [26,78]. This builds confidence and trust and a therapeutic relationship [65,76]. This can be seen schematically in Figure 4.

### 3.5. Feeling Safe

The concept of safety was predominant in our data; it derived from 16 papers (Table 3), covering all types of coercion. This indicates that feeling safe is important for people in order to be able to deal with the experience, regardless of the specific type of coercive measure. Feelings of safety develop in fear-free environments [79], where people are protected from destructive expressions of themselves and others [66,67,73,75].

Feeling safe is largely dependent on staff attitudes and behaviours. In a fear-free and safe environment, staff is experienced as trustworthy, fair and reliable [68,71]. Staff members balance service users’ needs with the constraints of acute wards [75]. They do not have custodial attitudes [68,79]; they are controlling without overreacting [66]. Persons subjected to coercion are able to assert themselves and to voice dissenting views without fear of further restrictions [79]. Mental health professionals adopt a tiered/discretionary approach to using restrictive measures [68] and keep them as short as possible, while also using alternatives [65,68,75].

Feeling safe is linked to consistent and reciprocal communication with staff before, during and after coercion. It is important that staff stays with the person throughout the coercive event. A sense of being carefully watched and being monitored for one’s own safety fosters this feeling. For example, in the study by Ezeobele et al., 2014 [67] participants valued it when a staff member was there, present and observing, without talking to someone else or engaging in other duties. Human contact between staff and the person subjected to coercion, such as talking or engaging in meaningful activities, fosters this feeling of safety [71,73,75]. Physical contact also provides comfort and resonates a caring and protecting attitude. In the words of a participant in Ling et al.’s (2015) [75] study:


*“Staff did the best thing, covered me with a blanket and gave me music and water too.”*
(p. 389)

Feeling safe is linked to the other two conditions for recovery from coercion. People feel safe when they are treated with dignity and respect and receive competent humane care by professional trustworthy staff members [68,71,75]. In a safe space, service users have an opportunity to reflect on their experiences and learn from them [70]. In this way, they can regain control and agency over their distressing experiences and mental state, which promotes empowerment. When coercion is experienced as a necessary measure, and not as punishment [73,79], people may use it as an opportunity to understand and work on their distress. Under the right circumstances, in a safe space, it might even be possible for people to accept coercion as a safety net, a last resort [73,76,79], when they cannot help themselves any more [83].

### 3.6. Being Empowered

Empowerment was addressed in 13 studies (Table 3) covering the whole range of coercive measures studied.

Good professional care allows people the highest possible level of empowerment, even under the most challenging circumstances [62]. Service users appreciate staff taking risks in giving them agency and control despite their legal status [26,63,77].

Having choices and being provided with options is important [68,77]. A participant in Fletcher et al.’s (2019) [68] study of locked doors said:


*“I probably look at it that you’ve got a choice, you could sit and watch telly if you want, you can sit out in the courtyard, but a lot of the time when you’re on a ward your choices are taken away from you, you know you’re medicated, you know you’re there sometimes under your, you don’t want to be there, so it’s about having that choice.”*
(p. 543)

In a study of involuntary hospitalisation [61], participants needed to feel that they still have some control and that the health care staff make no more decisions about them than necessary but instead focus on what is essential to their health and recovery.

Service users became empowered when they were facilitated to make sense of their mental distress, were provided with treatment options and were able to negotiate their treatment plan. In Bonner et al.’s study (2002) [63], some individuals regained some control when they were supported to make sense of their mental distress:


*“I’ve seen some of my records. It’s helped to look at the impression that they had of me at the time. It’s helped to build a better picture. The nurses have sat down with me and gone through my records with me. It feels better to look back with a better insight. You can see where you were going wrong. You can see where to make changes or where you’ve made changes”*
(p. 469)

In studies of CTOs [62,79,80,83], empowerment was supported when professionals listened and invited collaboration into decision making; thus, they were perceived as more likely to understand the patients’ issues and offer acceptable treatment regimes. In a participant’s words:


*“I think my CPN [Community Psychiatric Nurse] takes on board what I say she’s quite good, I can like test the waters with her and then we will think about it and not just on one single answer but look for a variety of avenues to follow.”*
[69] (p. 794)

Active involvement in decisions around one’s care promotes ownership; rather than being passive recipients of treatment, service users share “responsibility for adherence”, collaborating rather than complying [64].

### 3.7. Means to Achieve Recovery

Being in a facilitating environment, receiving appropriate information and having consistent reciprocal communication with staff are the means through which the conditions for recovery can be achieved.

### 3.8. Being in a Facilitating Environment

Being in a facilitating environment during coercion may strengthen the factors that support recovery from coercion. This was addressed in five studies (Table 3) about involuntary hospitalisation, locked doors, seclusion and restraint; this indicates the importance of the environment in forms of coercion that entail being restricted in a designated space against one’s will.

The studies agreed that a facilitating environment entails allowing physical freedom and encouraging meaningful activities [68,72,73,75,76]. Participants appreciated physical freedom to move inside and outside the ward and have access to leisure and therapeutic spaces. This allowed them to get fresh air, interact with others, receive visitors and engage in activities [68,72,73,75,77]. Meaningful activities help reduce boredom and restlessness, manage the “bad days”, reduce the desire to abscond and provide choices for spending the day [68]. They also provide opportunities for interaction with others. A participant reflected in a study of McGuinness [76]:


*“Within week 3 then em I began, I was mixed in with the other patients and I began attending different activities… they were well organised […] just doing different things from relaxation to solutions to wellness to eh how to deal with anxiety and panic attacks…so […] I’ve been just getting progressively better.”*
[76]

Experiencing physical freedom and engaging in activities helps to re-develop a sense of autonomy and being involved in valuable social roles [72].

The environment needs to be patient-friendly and humane, attending to people’s basic needs as well as their needs for comfort and safety. Kontio et al. (2012) captured how facilities during seclusion and restraint did not allow service users to fulfill their basic needs. Participants had many proposals related to turning the seclusion/restraint rooms into more humane environments, including the possibility of going to the toilet, having a nice meal, adequate furnishings, a window and a clock, which would make it possible to follow daily routines [73].

A facilitating environment includes opportunities for human contact, communication and support from staff and peers [68,76]. Furthermore, participants in Fletcher et al.’s (2019) [68] study stressed the need for wards to have peer support workers, who can provide support by listening and sharing their recovery story, providing advocacy and being a consistent support person from admission to discharge. Similarly, in Kontio et al.’s (2012) [73] study participants stressed the value of talking about their experience with an external evaluator, a patient representative.

The architecture of the environment includes spaces for physical activities and social interaction as well as spaces for calming, comfort and privacy. In Ling et al.’s (2015) [75] study, participants preferred spaces and activities that help in escaping the crowded and busy unit environment, such as private bedrooms or comfort rooms, as well as comfort and sensory interventions by staff before and during restraint.

A facilitating environment fosters all three conditions for recovery, as can be seen in Figure 5. It provides the context for treating service users with dignity and respect, helping them feel safe, fostering their empowerment. When the ward environment is facilitating, the setting is therapeutic and caring, it brings hope and healing instead of custody. The environment could reframe the entire meaning of coercion, from an abusive and degrading experience to the turning point of a renewing/reappraisal process.

### 3.9. Receiving Appropriate Information

There are 13 papers that addressed the importance of receiving appropriate information regarding the coercive event (Table 3). Providing appropriate information might help people understand the reasons for exercising coercion on them [71]. In a situation where the person has no control, the minimum staff can do is not hide what is being done to them and why [68].

Appropriate information is clear, adequate and realistic information about the ward rules and routines, the reasons for the coercive intervention, the process of the intervention and the possible consequences for the person [61,66,68,75,81]. Information must be patient-centred, tailored to the person’s needs and history [61,66,75], ensuring that the patient understands what is happening. It must be given at the right time, not too soon and not too late, several times before, during and after the event, in oral and written forms. It must be delivered in trauma-sensitive language, acknowledging the dilemmas for both staff and service users [66,73,74]. A staff participant in Banks et al.’s (2016) [62] study explained:


*“[For] some people it’s going back two or three times and discussing it with them, the lady who we see twice a day, it was discussing it twice a day because she really didn’t understand and so you are constantly having to remind her and discuss it with her.”*
(p. 185)

Being given sufficient information gives people subjected to coercion a sense of being treated as a person, acknowledged and respected [81]. Being provided with information is experienced as caring and helpful [66]; as such, it forms the basis for a trusting relationship with staff [75,81]. It also provides a sense of understanding about what is happening, thus fostering safety [74] and a sense of control over the situation. It mobilises the person’s cognitive processes, enabling them to process the experience [61] and provides a sense of involvement in one’s own care, which nurtures autonomy and fosters empowerment [73]. Figure 6 presents schematically how receiving appropriate information fosters the conditions for recovery from coercion.

However, appropriate information seems to be rarely given. In many studies [65,66,73,75,79,83], participants reported that nobody talked to them. Reportedly, staff either do not talk to the patients at all about the coercive measure or they simply announce decisions regarding treatment without explaining or discussing them with the service users. This leaves people feeling unacknowledged, treated as a problem and not as a person [69] and powerless [65] (Chambers et al., 2014). This may be, at least partly, due to the specific circumstances related to decision-making processes during coercion. The process must be promptly executed, against the person’s will, while the person is in a psychological crisis and the staff does not have the time. It could also be due to the paternalistic approach prevalent in many mental health facilities, whereby staff are responsible for making decisions and imposing treatment for the patients’ benefit. This means that mental health professionals have “*no investment in explaining it clearly*” (Banks et al., 2016, p. 185) [62] and results in what participants in Pridham et al.’s (2018) [79] study described as “*rubber stamping*” (p. 125), where giving information is just something that must be done.

### 3.10. Engaging in Consistent Reciprocal Communication

Recovery from coercion presupposes consistent and reciprocal communication with staff who listen and engage. The significance of communication with staff was addressed in eight studies (Table 3).

Consistent reciprocal communication is established when staff are perceived as friendly and willing to listen [66,76,77]. In Chien et al.’s (2005) [66] study of physical restraint, it was noted that:


*“The nurse came to my bedside and told me who she was and said she would be available nearby for my requests during the shift. She also came to talk to me from time to time. This showed that she cared about me and she let me talk with her if I needed to.”*
(pp. 82–83)

Feeling connected requires staff to do their job properly but also to engage with the patient as a person, listening and attending to their thoughts and needs [77].

Communication with staff builds confidence and trust and a therapeutic relationship [75,77]. Patients feel that they are listened to and cared for and treated as a human being, preserving their dignity and self-respect [66,73,74]. In a participant’s words:


*“I need a human being beside me. I want to talk about my fears with the physician and nurse. I like to have a connection to them, now they are in a hurry all the time.”*
(Kontio et al., 2012 [73], p. 21)

When having the feeling of connectedness, service users feel less vulnerable and thus safer [74,75]. They retain a degree of agency and control, even during coercion, thus feeling empowered. This enables them to make sense of their distress and accept treatment, which fosters their recovery [77]. As a participant said:


*“She [nurse] was talking to me as though she believed what was going on in my thoughts … she understood where I was coming from … asking me questions that were trying to make me think introspectively.”*
(McGuinness et al.’s (2018) [77], p. 504)

The ways in which engaging in reciprocal communication is linked to the three conditions for recovery can be seen in Figure 7.

### 3.11. Strategies to Achieve Recovery

People subjected to coercion seem to employ strategies for managing coercion to achieve recovery, both during the coercive event, to minimise coercion and its impact, and after coercion, to process the experience. However, these strategies can only be employed if the conditions and means outlined above are in place. Only a few studies mentioned strategies that people utilise to deal with coercion (Table 3), indicating, on the one hand, the overwhelming and traumatic character of coercion and, on the other, the lack of opportunities for people to devise ways to manage it and mitigate its impact.

### 3.12. During Coercion—Strategies to Minimise Coercion and Its Impact

When being coerced, patients may resist restriction and seek some level of agency and control over the situation. In the study of involuntary hospitalisation by McGuinness et al. (2018) [77], participants seek agency and control through confronting professionals; they verbally assert themselves, physically protest and do not follow professionals’ directions. When some level of choice is provided and agency is allowed by professionals, it encourages a sense of autonomy, even during the restraining event. A level of control might be achievable when patients have a choice to participate in activities during involuntary hospitalisation [68], when they are supported to make sense of their mental distress [63] or when they are allowed to negotiate treatment options [79].

Appearing compliant with staff expectations is frequently used as a coping strategy. In the study by Hughes et al. (2009) [72], involuntarily detained participants discuss deliberately appearing to comply with staff expectations in the hope of earlier discharge. Similarly, in Gault et al.’s (2013) [69] study of compulsory medication some service users foresee professional expectations and “*play the game*” (p. 508); they appear compliant by regulating their medication themselves, while concealing it from professionals. According to a participant:


*“I’m good at being compliant, my friend got into trouble, but I didn’t argue.”*
(Gault et al., 2013 [69], p. 793)

Another strategy that participants use is resignation, just following the rules, doing what they are told, to get through the situation and not get into trouble. In McGuinness et al.’s studies [76,77], some individuals who saw no benefit from being involuntarily admitted conformed to the system by keeping their head down, saying nothing [77], resigned to taking medication just in order to get out [76].

If the conditions for recovery are present, people might recognise them and use the coercive situation as an opportunity for recovery, through making use of the safe space and communication with staff, therapeutic relationships and the treatment offered for the benefit of their mental health. In Pridham et al.’s (2018) [79] study, service users and staff argue that CTOs may enable recovery through medication compliance and lessening symptoms; in this way, clients regain control over their lives and mental health. According to the study by McGuinness et al. (2018) [77], being provided with treatment approaches during involuntary hospitalisation helped patients understand and gain perspective of what is happening to them, allowing the possibility of recovery:


*“…when I was in [names hospital] I really sort of faced up to my issues…I think I had a lot of built-up anger, resentment, regrets and other things that had been below the surface for many years and I hadn’t sort of dealt with things […] now I understand more about myself.”*
[77] (pp. 504–505)

### 3.13. After Coercion—Processing the Experience

Processing the experience of coercive measures afterwards is depicted as a lonely feeling, whereby people must help themselves to deal with the traumatic experience and its later impact. The experience created an urge to talk to someone, even after a long time, but people felt that others were not interested or would not understand [71]. Some were able to talk to someone outside the hospital, e.g., with a GP or an external evaluator [73]. Whether it was suffered in silence or shared with others, the experience could fade over time, making time a restorative factor. Some people ended up learning to live with the experience rather than assimilating it in their lives [71].

Both service users and staff recognised the importance of active engagement with the persons subjected to coercion after the event [63,75,77]. It did not have to be a major thing, just a little talk over a cup of tea (Bonner et al., 2002, p. 470). This would give an opportunity for people to retell the story and explain their experience, where staff listened. This kind of debriefing was seen as an opportunity to regain trust [75]. Post-incident review (PIR) was only mentioned in three papers [70,73,74]. PIR was suggested to be an arena for reflections amongst all involved parties, an opportunity to talk about the event and the reasons for it. This could render PIR a learning experience, with the potential to improve the quality of care based on knowledge about other perspectives and solutions, increased professional and ethical awareness and increased care providers’ emotional and relational processing [70]. Despite acknowledging its importance, PIR was rarely done in practice. People who had experienced coercion thus ended up being solely responsible for processing the experience by means of their own strategies, with no help from professionals.

## 4. Discussion

Research on coercion in mental health care has focused on the nature of coercion, the means to prevent coercion and the adverse effects of coercion, with very limited evidence on how one can recover from coercion. Moreover, although there are guidelines on how to prevent coercion [84], no official guidance exists on how to facilitate recovery from coercion. This is the first international study to systematically review and integrate qualitative research in order to increase our understanding of subjective experiences of recovery from being exposed to coercive measures of persons with mental health conditions, health care providers and relatives.

The review suggests that recovery from coercion depends on three interwoven and complex sets of factors: conditions for recovery from coercion; means to achieve these conditions; and strategies for dealing with the impact of coercion. These parallel different aspects of organisations: conditions are related to the organisational culture, means to professional practices and strategies to the service users’ coping strategies. According to organisational management theory [85], organisational culture is a dynamic phenomenon that surrounds the organisation at all times, being constantly enacted and created by interactions within it, and it is also shaped by leadership behaviour and a set of structures, routines, rules and norms that guide and constrain behaviour. Adopting such a dynamic view of organisational processes allows us to examine how these factors influence each other and how all parties may or may not contribute to achieving recovery from being subjected to coercion.

Central to the qualitative findings seemed to be experiencing dignity and respect, which functions as a pre-condition for recovering from coercion. Service users retain their dignity and feel that they are treated with respect when they feel acknowledged as a person in distress and treated as a human being and not as a diagnosis, a sick patient or a problem. This is in line with Verbeke et al. (2019) [86], who found that patients, when treated with dignity and respect, can cope better with coercion, understand better why coercion was used and recover more easily from the event. This is also supported by accounts from service users and theoretical work [16].

In line with Lindgren et al. (2019) [87], we found two other primary conditions for overcoming the negative consequences of coercion: feeling safe and being empowered. These three conditions are, of course, interlinked. Lindgren (2019) [87] found that safety could be fostered by supporting autonomy and promoting a sheltered experience during isolation. Feeling safe may be possible when one is treated with dignity and respect. When people feel they are treated as a disease or a danger instead of a person, methods to induce safety will be experienced as invasive controlling mechanisms. Conversely, a facilitating empathic environment that encourages freedom and empowerment will make it easier for staff to treat patients with respect, even when in crisis. Empathic environments make staff more flexible to support empowerment [88]. These three interwoven conditions could be understood as a feedback loop [89]. When coercion is used in a context of dignity and respect, people subjected to it feel that it is their symptoms and problems that are kept in control, and not them as a person. The patient is still there as a human being, who even during a psychotic crisis [90] is able to reflect on what is happening and can have some sense of control over the process. If, by contrast, patients feel that staff use rules and norms to guide and constrain their behaviour, and do not start from a basic attitude of solidarity and fundamental sympathy, coercion can then be interpreted as central to their identity and will be traumatic [23].

According to Verbeke et al. (2019) [86], recovery from coercion does not depend on specific interventions or techniques but much more on the relationship between patient, staff and ward. This review demonstrates how the culture and environment of the ward are influenced by practices and routines, such as how rules and structures are performed in the communication between staff and patients, as well as how and if information regarding the reasons for specific interventions is given to them. These practices are paramount and can either facilitate or hinder recovery from coercion. This is what Sjöström (2006) calls the coercive context; coercion is not something dichotomous, a specific act that happens or does not happen, but something that resides within a broader context and permeates it [91]. Coercion is experienced mostly negatively within a power relationship, where patients are reduced to their problems and not seen as human beings. When there is respect and dignity during coercion, patients have some kind of control, and the therapeutic alliance is kept, then the experience is less negative [19,86,87]. Wards that engage in more humane care, with a focus on respect, open communication, debriefing and empowering patients, are less coercive (e.g., Safewards and Sixcore strategies) [92]. This might be because there is less aggression within contexts that are less controlling and because caregivers who are emotionally closer to their patients see them more easily as unique human beings and use less coercion [93]. We could, therefore, hypothesise that a patient will recover from a coercive incident more easily when this does not take place within a broader coercive context.

We could state, on the basis of our findings, that recovery from coercion is a systemic and personal process, which is influenced by the staff’s ability to create surroundings that facilitate sustained humane empathic contact and promote a sense of safety. Surroundings that reverberate negative values, such as ignoring the patient’s thoughts and needs [94], can result in service users adopting non-helpful strategies that this study identified; they may protest, comply or just follow the rules, without really believing in them. This resignation is negative [19] and can increase patients’ feelings of disempowerment and amplify their loss of self-esteem, which is already under pressure because of their symptoms and the coercion itself. Our study showed that people subjected to coercion are rarely supported to make sense of and manage these traumatic experiences by professionals, relatives or peers. Instead, they are customarily left to manage on their own during the coercive event and suffer in silence after it. We found that people do employ coping strategies, during the coercive event, in order to minimise the event itself and its impact, and after the event, to process the experience, using whatever resources they have available to them. This is where the ward environment and staff are crucial; if they are facilitative, service users can employ more effective strategies and, if they are not, then the range of strategies will be limited, as well as their effectiveness, and this would hinder their chances of recovery.

The 23 papers included in the meta-ethnography covered all types of formal coercion, from the most intrusive ones, like seclusion and restraint, to the least intrusive, like CTOs. It may be assumed that the degree of restriction and intrusiveness of each type of coercive practice would have differential impact on those subjected to it. It is noteworthy, however, that no major differences were found between these types of coercion regarding what people find distressing or facilitative, indicating that it is the experience of coercion per se, rather than the type or degree, that has the negative impact. This review included studies of exposure to formal coercion, but people might not differentiate between formal and non-formal coercion [16]. Indeed, studies of informal coercion [6,9,68] arrive at similar findings. This further supports the argument that coercion is experienced in similar ways, regardless of the specific type, and that, correspondingly, the factors that facilitate or hinder recovery may be similar.

This review overwhelmingly highlights the importance of the mental health service culture and the professional practices for promoting or hindering recovery of service users, pointing to the need for reforming mental health services in a recovery-oriented direction. This can be challenging because the internal structures, rules and regimes can counteract human care [95]. Organisations are living organisms, where the management system influences the behaviour of its members and vice versa. It is difficult to implement recovery-oriented practices within traditional cost-conscious mental health service environments, where treatment is defined narrowly, promoting a biomedical individualistic approach to mental distress. Within this traditional approach, there can be a reluctance to shift from the biomedical causal models to a holistic biopsychosocial model of mental distress that is recovery oriented [96].

### Strengths and Limitations/Reflexivity

This is the first published meta-synthesis of studies on how people recover from being subjected or exposed to coercive measures in mental health care. Given the novelty of the topic, utilising a meta-ethnographic approach allowed us to work interpretatively and in depth with the limited studies available and to arrive at a model of the factors and processes that facilitate recovery from coercion.

The author group consisted of clinicians, academics and researchers from all the mental health disciplines, representing six European countries. The broad range of backgrounds, skills, expertise and experience of the authorial team were put to use in the continuous collaborative process of working on this project; this is a significant strength of this study. All authors are members of the Fostering and Strengthening Approaches to Reducing Coercion in European Mental Health Services (FOSTREN) network, sharing a commitment to understanding the impact of coercion on those subjected to it and exploring alternatives that would reduce coercion and/or mitigate its impact. Though, there was no expert by experience in the author group, which is a weakness of the study.

Another limitation may be that there is no specific guidance for judging the quality of a meta-ethnography; however, the transparent reporting done in this study and the feedback from the author group enhanced its trustworthiness.

The review included only studies of formal coercion. Although there are indications in the literature that informal coercion is experienced in similar ways to formal coercive events, and we might assume that the factors and processes that we identified might apply to recovery from informal coercion, we cannot deduce that this is the case on the basis of our data. Also, we focused on adult mental health settings, excluding child and adolescent mental health, forensic, addiction and intellectual disability settings. This limits the generalisability of our findings; we can only partially generalise on grounds of similarities between the settings.

## 5. Conclusions

Despite the significant growth of research related to human rights in mental health care, there is limited knowledge related to recovery from being subjected to coercive measures. Through synthesising the findings of qualitative studies examining views of different groups of persons involved in coercive events on what may help manage the traumatic impact of coercion, this study produced a map of conditions, means and strategies that may be used to facilitate recovery from coercion in mental health care.

Recovery from coercion is possible when mental health care settings offer recovery-oriented environments and mental health professionals employ recovery-oriented practices that would empower service users to develop strategies for managing their mental distress as well as their experiences in mental health care in a way that minimises traumatisation and fosters their recovery. The basic requirements for a patient to recover positively from such an event seem to be receiving treatment based on dignity and respect, feeling safe and not feeling diminished by the treatment received but, if possible, empowered. Humane, caring and respectful treatment that considers the patient as a whole person with emotions, thoughts, sensitivities and affections, and not just an acute and dangerous condition to be treated, can ensure improvement in the qualities of care, patient and staff satisfaction and sense of safety, in a calm and recovery-oriented work environment.

The model of factors and processes that facilitate recovery from coercion that we generated can be used to raise an awareness among all involved and guide new practices and policies, regulations and research.

## Figures and Tables

**Figure 1 healthcare-12-00628-f001:**
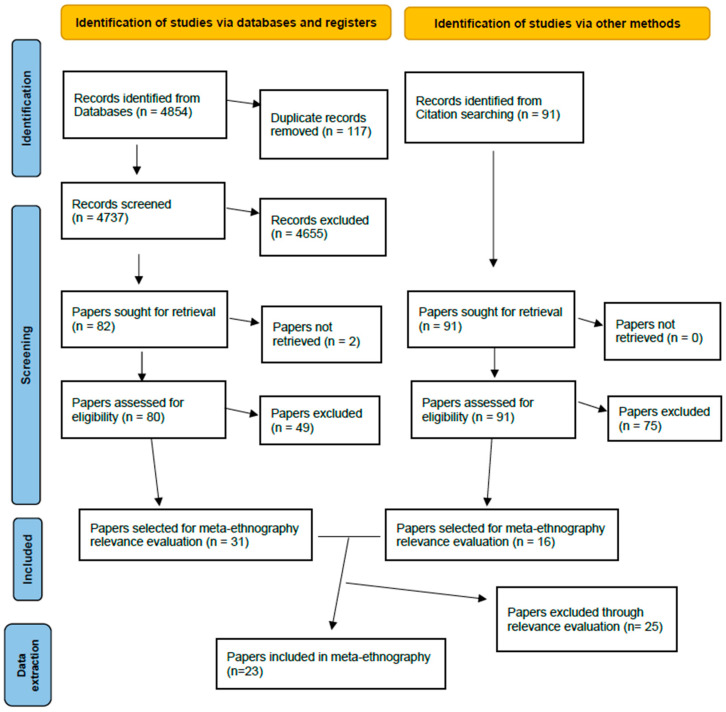
PRISMA flow chart.

**Figure 2 healthcare-12-00628-f002:**
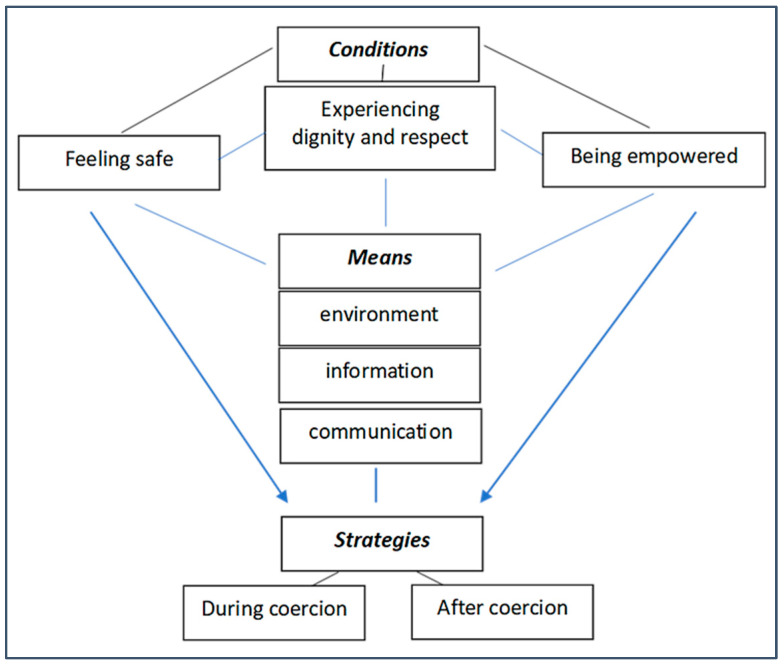
Synthesis of factors and processes involved in recovery from coercion.

**Figure 3 healthcare-12-00628-f003:**
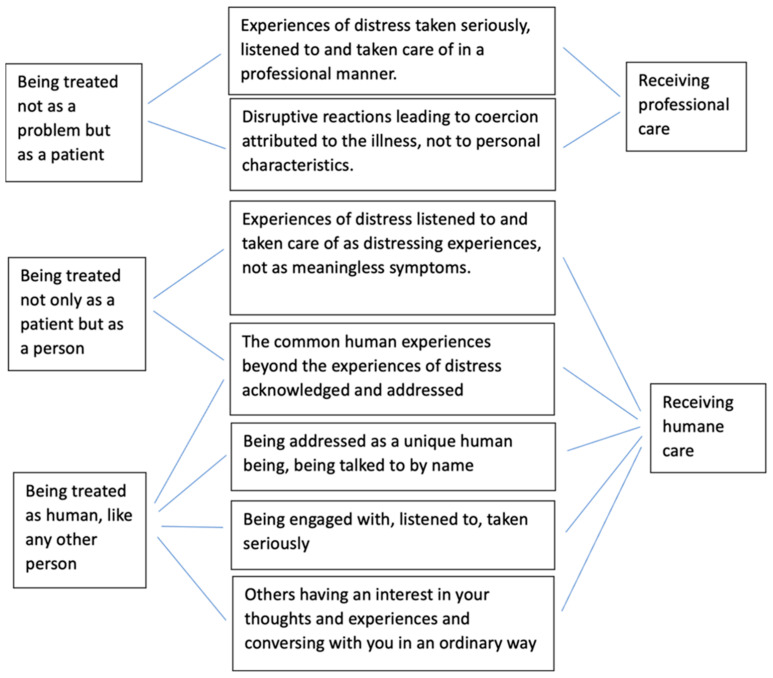
Components of being treated as a person in distress.

**Figure 4 healthcare-12-00628-f004:**
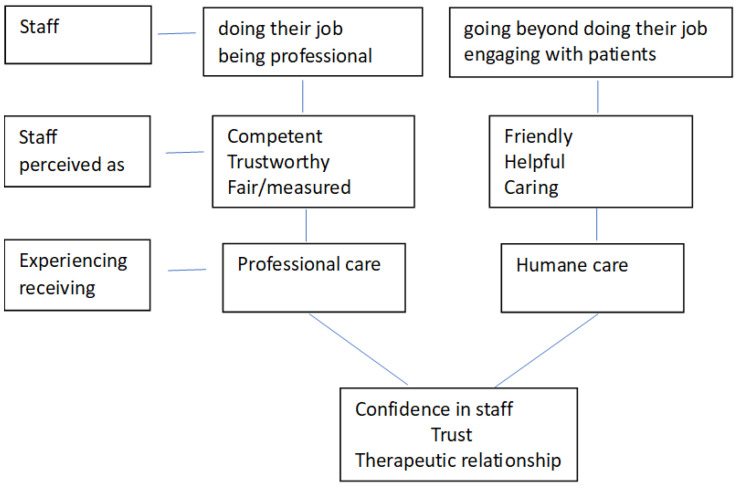
Components of care promoting dignity and respect.

**Figure 5 healthcare-12-00628-f005:**
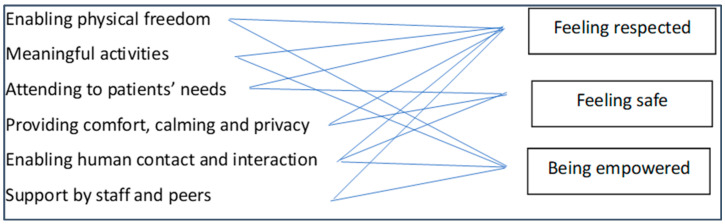
A facilitating environment fosters the conditions for recovery.

**Figure 6 healthcare-12-00628-f006:**
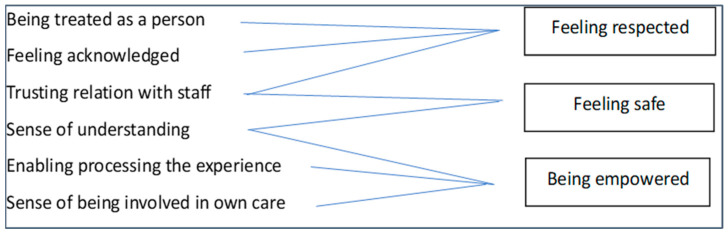
Receiving appropriate information fosters the conditions for recovery.

**Figure 7 healthcare-12-00628-f007:**
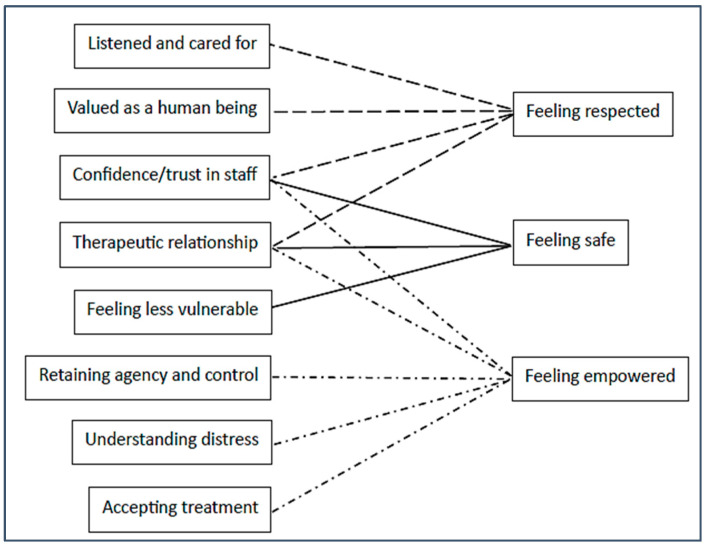
Consistent reciprocal communication fosters the conditions for recovery.

**Table 1 healthcare-12-00628-t001:** PubMed search string.

PubMed ((((((“Coercion”[Mesh]) OR “Restraint, Physical”[Mesh]) OR (coerci*[Title/Abstract] OR involuntary[Title/Abstract] OR restraint[Title/Abstract] OR seclusi*[Title/Abstract] OR compuls*[Title/Abstract] OR force*[Title/Abstract] OR pasung[Title/Abstract] OR confinement[Title/Abstract])) NOT (“compulsive disorder”[Title/Abstract])) AND ((((“Mental Health Recovery”[Mesh]) OR (“Rehabilitation/growth and development”[Mesh] OR “Rehabilitation/psychology”[Mesh])) OR (recover*[Title/Abstract] OR connectedness[Title/Abstract] OR hope[Title/Abstract] OR identit*[Title/Abstract] OR meaning*[Title/Abstract] OR empower*[Title/Abstract] OR self-directed[Title/Abstract] OR “quality of life”[Title/Abstract]))) AND (((((“Psychiatry”[Mesh]) OR “Mental Disorders”[Mesh]) OR “Hospitals, Psychiatric”[Mesh]) OR “Mental Health Services”[Mesh]) OR (psychiatr*[Title/Abstract] OR mental illness*[Title/Abstract] OR mental disorder*[Title/Abstract]))) AND ((((((“Qualitative Research”[Mesh]) OR “Hermeneutics”[Mesh]) OR “Anthropology, Cultural”[Mesh]) OR (“Interview” [Publication Type] OR “Interview, Psychological”[Mesh])) OR (“Narration”[Mesh] OR “Personal Narrative” [Publication Type])) OR (narrative* or “lived experience*”[Title/Abstract] or qualitative or “thematic analys*”[Title/Abstract] or hermeneutic or interview* or interpret* or phenomenol* or transcr* or “focus group*”[Title/Abstract] or “grounded theory”[Title/Abstract] or open-ended or perspective* or experienc* or first-person* or etnograph*))

**Table 2 healthcare-12-00628-t002:** Characteristics of included studies.

Authors, Date Country	Type of Coercion	Aim/Focus	Setting	Participants	Data Collection Method	Analysis Method
Andreasson and Skärsäter, 2012 Sweden [61]	inv.hospital.	experiences of compulsory treatment during acute psychosis	acute psychiatric units	12 service users	interviews	phenomenographic
Banks et al., 2016 UK [62]	CTO	experiences of CTO within a policy context of person-centred care	mental health trust	72 service users, carers and staff (30 staff)	interviews	thematic
Bonner et al., 2002 UK [63]	physical restraint	experiences and effects of physical restraint	acute psychiatric units	service users and staff involved in 6 physical restraint incidents	interviews	qualitative
Canvin, et al., 2002 UK [64]	CTO	perceptions and experiences of living with CTO	mental health trusts and social service departments	20 service users	interviews	qualitative
Chambers et al., 2014 UK [65]	inv.hospital.	experiences of dignity and respect during involuntary hospitalisation	acute psychiatric units	19 service users	interviews	thematic
Chien et al., 2005 China [66]	physical restraint	experiences of first encounter with physical restraint	acute psychiatric units	30 service users	interviews	content
Ezeobele et al., 2014 USA [67]	seclusion	lived experience of seclusion	acute psychiatric units	20 service users	interviews	qualitative
Fletcher et al., 2019 Australia [68]	locked doors	test the acceptability of recommendations for providing least restrictive, recovery-oriented practices	regional and urban locations	9 service users 9 carers 7 staff	forum discussions	thematic
Gault et al., 2013 UK [69]	compulsory medication	perceptions of medication adherence	various community locations	18 service users 6 carers	interviews and focus groups	qualitative
Hammervold et al., 2020 Norway [70]	physical and mechanical restraint	staff experiences and considerations regarding post-incident reviews (PIRs) after restraint	acute psychiatric units	9 nurses, 3 social educators, 4 doctors/psychiatrics, 3 psychologists	interviews	qualitative content
Hoekstra et al., 2004 Netherlands [71]	seclusion	experiences and effects of seclusion	acute psychiatric units	7 service users	interviews	grounded theory
Hughes et al., 2009 UK [72]	inv.hospital.	perceptions of the impact of involuntary hospitalisation on self, relationships and recovery	acute psychiatric units	12 service users	interviews	thematic
Kontio et al., 2012 Finland [73]	seclusion and restraint	experiences and suggestions for improvement of seclusion/restraint	acute psychiatric units	30 service users	interviews	qualitative content
Lanthén et al., 2015 Sweden [74]	mechanical restraint	experience of mechanical restraint and care received	acute psychiatric units	10 service users	interviews	qualitative
Ling et al., 2015 Canada [75]	seclusion and restraint	experiences before, during and after a restraint event	acute psychiatric units	post-restraint event debrief forms for 55 service users	audits on inpatient charts	thematic
McGuinness et al., 2013 Ireland [76]	inv.hospital.	experiences of involuntary hospitalisation	acute psychiatric units	6 service users	interviews	IPA
McGuinness et al., 2018 Ireland [77]	inv.hospital.	experiences of involuntary hospitalisation	acute psychiatric units	50 service users	interviews	grounded theory
Murphy et al., 2017 Ireland [26]	inv.hospital.	experiences of involuntary hospitalisation	acute psychiatric units	50 service users	interviews	thematic
Olofsson and Jacobsson, 2001 Sweden [78]	inv.hospital.	experiences of coercion during involuntary hospitalisation	acute psychiatric units	18 service users	interviews	qualitative content
Pridham et al., 2018 Canada [79]	CTO	experiences of and perspectives on CTO	community mental health locations	9 service users 6 carers 12 staff	interviews	thematic
Stroud et al., 2015 UK [80]	CTO	experiences of and perspectives on CTO	mental health trust	21 service users 7 carers 35 staff 9 service providers	interviews	thematic
Wyder et al., 2015 Australia [81]	inv.hospital.	experiences of relationship with staff during involuntary hospitalisation	acute psychiatric units	25 service users	interviews	qualitative
Wyder et al., 2016 Australia [82]	inv.hospital.	tensions between empowerment and control during involuntary hospitalisation	acute psychiatric units	25 service users	interviews	qualitative

**Table 3 healthcare-12-00628-t003:** Frequency of categories per study included.

		Conditions			Means			Strategies	
Authors, Date	Type of Coercion	Experiencing Dignity and Respect	Feeling Safe	Being Empowered	Being in a Facilitating Environment	Receiving Appropriate Information	Consistent Reciprocal Communication	Strategies during Coercion	Strategies after Coercion
Andreasson and Skärsäter, 2012 [61]	inv.hospital.	x		x		x			
Banks et al., 2016 [62]	CTO			x		x			
Bonner et al., 2002 [63]	physical restraint	x		x				x	x
Canvin et al., 2002 [64]	CTO		xx	x					
Chambers et al., 2014 [65]	inv.hospital.	x				x			
Chien et al., 2005 [66]	physical restraint	x	xx			x	x		
Ezeobele et al., 2014 [67]	seclusion	x	xxx						
Fletcher et al., 2019 [68]	locked doors		xxxxx	xx	xxxx	x		xx	
Gault et al., 2013 [69]	compulsory medication	xx	xxx	x		x	x	xxxx	
Hammervold et al., 2020 [70]	physical and mechanical restraint		x						x
Hoekstra et al., 2004 [71]	seclusion		xxxxxx			x			xxx
Hughes et al., 2009 [72]	inv.hospital.	x	xxx	xx	xxxx		xx	xx	
Kontio et al., 2012 [73]	seclusion and restraint	x	xxxx		xxxx	x	x		x
Lanthén et al., 2015 [74]	mechanical restraint		xxxxx			x	x		x
Ling et al., 2015 [75]	seclusion and restraint		xxxxx	x	xxx	xx	x		x
McGuinness et al., 2013 [76]	inv.hospital.	x	xx		x		xxx	xxxxxx	
McGuinness et al., 2018 [77]	inv.hospital.		x	xxx			xx	xxxxxx	x
Murphy et al., 2017 [26]	inv.hospital.	x							
Olofsson and Jacobsson, 2001 [78]	inv.hospital.	xx		x					
Pridham et al., 2018 [79]	CTO		xxxx	xx		x		xxx	
Stroud et al., 2015 [80]	CTO		x	x					
Wyder et al., 2015 [81]	inv.hospital.	xx	x	x		x			
Wyder et al., 2016 [82]	inv.hospital.		x			x			

Note: x stands for each time a key concept was extracted from a paper.

## Data Availability

Data sharing is not applicable to this article as no new data were created or analysed in this study.

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
