# Peer review of "Factors and Processes Facilitating Recovery from Coercion in Mental Health Services—A Meta-Ethnography"

_healthcare, 2024, doi:10.3390/healthcare12060628_

Round 1

Reviewer 1 Report

Comments and Suggestions for Authors

-  The subject of this study is exciting primarily because it deals with phenomena that involve an obvious degree of social and professional desirability. It is a pity that the authors do not talk about social and professional desirability.

- I must note the lack of a literature review chapter. As far as I can tell, the "Introduction" chapter is actually a mix between introduction and literature review. I think there should be two separate chapters.

- The authors describe in subchapter 2.1.2 the search strategy. This description is clear enough, so I don't understand why Table 1 (row 152) is needed. In my opinion this table is extra and can be omitted.

- In relation to the application of the meta-ethnography method, I think that Step 4 - Determining how the studies are related (row 208) has not been emphasised enough. The relationships between the key concepts from the different papers need to be explained better. Concepts has to be explain and do not only describe.

- Authors show the conditions for recovery grouped in three central categories: experiencing dignity and respect, feeling safe and being empowered. This operationalisation of the conditions for recovery is necessary and welcome. However, there are two points I need to make:

1.            The terms respect and dignity are vague, difficult to understand. That is why they must be operationalised. I suggest that the authors insist on explanations of what dignity and respect mean in the context of treatment of people with mental health problems.

2.            I think an important element is missing from the scheme of interpretation of processes facilitating recovery from coercion. This is the degree of understanding and acceptance by the patient, which translates into the efforts of medical staff to ensure that the patient understands what is happening to him and that he is convinced that coercive measures are part of the treatment and that they are absolutely necessary.

Author Response

Reviewer 1

Comment

The subject of this study is exciting primarily because it deals with phenomena that involve an obvious degree of social and professional desirability. It is a pity that the authors do not talk about social and professional desirability.

Response

Thank you for this remark. Social and professional desirability was not a topic that we identified in the studies that formed our corpus, and this is why we do not mention it explicitly. We mention professional wishes, whenever we encountered them in our material, for example, in section 3.5 their desire to “…balance service users’ needs with the constraints of acute wards [75], or that they “do not have custodial attitudes [68,79]; they are controlling without overreacting [66]…”. Examples of social desirability might include references to having a talk over a cup of tea in section 3.13.

Comment

I must note the lack of a literature review chapter. As far as I can tell, the "Introduction" chapter is actually a mix between introduction and literature review. I think there should be two separate chapters.

Response

Thank you for this comment. It is customary and accepted in journal articles that there are no separate introduction and literature review sections, and that the literature review section is called introduction. This is what we have followed in this paper.

Comment

The authors describe in subchapter 2.1.2 the search strategy. This description is clear enough, so I don't understand why Table 1 (row 152) is needed. In my opinion this table is extra and can be omitted.

Response

Indeed, the table presents a very detailed description of the search, that was presented in the text. However, we think that it is important to show that we did a systematic literature search, as sometimes meta-ethnographic studies utilize only purposeful sampling.

Comment

In relation to the application of the meta-ethnography method, I think that Step 4 - Determining how the studies are related (row 208) has not been emphasised enough. The relationships between the key concepts from the different papers need to be explained better. Concepts has to be explain and do not only describe.

Response

We have added a sentence in that step: and compared the concepts found in each study with each other, e.g. being in a facilitating environment. It will be too much to explain in detail the steps in the method section, as it also is explained in the figures in the result section, which we have done in relation to every concept.

Comment

Authors show the conditions for recovery grouped in three central categories: experiencing dignity and respect, feeling safe and being empowered. This operationalisation of the conditions for recovery is necessary and welcome. However, there are two points I need to make:

1] The terms respect and dignity are vague, difficult to understand. That is why they must be operationalised. I suggest that the authors insist on explanations of what dignity and respect mean in the context of treatment of people with mental health problems.

Response

Yes, dignity and respect are the main concepts that emerged from our analysis, and we agree that they are difficult to understand. This is why we have dedicated a whole section, Section 3.4., where we describe the concepts in detail, and we added two figures, Figure 3 and Figure 4, that attempt to operationalize the concepts and portray their relation to other concepts that we find in our material. We have also now added a phrase in the text of Section 3.4., taken from Figure 3, that specifies that being treated with respect and dignity is being talked to by name.

Comment

  1. I think an important element is missing from the scheme of interpretation of processes facilitating recovery from coercion. This is the degree of understanding and acceptance by the patient, which translates into the efforts of medical staff to ensure that the patient understands what is happening to him and that he is convinced that coercive measures are part of the treatment and that they are absolutely necessary.

Response

Thank you for this remark. We added a sentence in relation to this in Section 3.9 about receiving appropriate information (lines 435-436).

Reviewer 2 Report

Comments and Suggestions for Authors

First of all, we find the subject of the paper very interesting and intriguing and we acknowledge the efforts of the authors in providing an extensive introduction through which the different meanings and consequences of coercion are discussed. The authors also provide comparative studies, between different cultures and legislations, since the measures of coercion may differ significantly based on these two aspects. 

The second chapter presents the materials and methods. It seems appropriate to describe in a few words what the Noblit and Hare framework is, in an attempt to give all readers a chance to comprehend better the scenario on which the research was based.

The following sections describe every step that was taken for the selection of the papers that were later analyzed by the authors. The PRISMA flowchart the authors use provides a summary of the long process of selection of the papers. However, it is needed to emphasize the exact criteria upon which the selection was done. The authors provide the keywords used for the discovery of relevant papers (rows 145-148), and the whole process is described in detail (rows 154-185), but with no emphasis on the criteria that were used either for selecting or for dismissing a specific article. 

Table 2 presents the main features of the remaining articles. 

In row 217 the authors state they determined the studies appropriate for their questions, but up to this point no reference about what the actual questions of their study is made from a methodological point of view. We feel that before starting to describe the whole selection process, the authors should have a section dedicated to their study's objectives and/or research questions.

The whole section of findings addresses the studies and the interpretation the authors made regarding the variables they showed interest in. The whole section is very adequately referenced, and the authors provide meaningful insight about each category. This section of the study is comprehensive and extensive. 

The authors discuss all their findings emphasizing the categories on which recovery from coercion depends. From organizational elements to personal traits, from human relationships and interactions to the recovery process itself, the authors describe extensively all the factors identified in the scientific literature. 

The section regarding the strengths and limits of this research is also well documented, the authors acknowledge elements that can be improved in future research.

The conclusion section suggests the importance of the findings for the well-being of patients who benefit from healthcare settings. We find the results are crucial for the possibility of managing mental well-being and feel the need to congratulate the authors for their extraordinary work. 

Author Response

Reviewer 2

Comment

First of all, we find the subject of the paper very interesting and intriguing and we acknowledge the efforts of the authors in providing an extensive introduction through which the different meanings and consequences of coercion are discussed. The authors also provide comparative studies, between different cultures and legislations, since the measures of coercion may differ significantly based on these two aspects. 

Response

Thank you

Comment

The second chapter presents the materials and methods. It seems appropriate to describe in a few words what the Noblit and Hare framework is, in an attempt to give all readers a chance to comprehend better the scenario on which the research was based.

Response

We have added a few words of what the Noblit and Hare framework is, thank you for pointing this out.

Comment

The following sections describe every step that was taken for the selection of the papers that were later analyzed by the authors. The PRISMA flowchart the authors use provides a summary of the long process of selection of the papers. However, it is needed to emphasize the exact criteria upon which the selection was done. The authors provide the keywords used for the discovery of relevant papers (rows 145-148), and the whole process is described in detail (rows 154-185), but with no emphasis on the criteria that were used either for selecting or for dismissing a specific article. 

Response

The exact criteria upon which the selection was done is described in 2.1.2.(describing what is relevant): “with the aim of the review, the search terms included variations of terms related to a) coercion, b) mental health/psychiatry, c) recovery and a) qualitative methodology”. After this we exemplify the search string. We also give detailed information about inclusion and exclusion criteria, in the same section: “Only empirical studies about formal coercion were included, excluding papers about informal coercion, minors, elders, forensic settings, persons with addictions and mentally impaired persons. Qualitative studies, mixed method studies, and research case studies were included; surveys, quantitative studies, opinion papers, and clinical report studies were excluded”. We think this explains it adequately.

Comment

Table 2 presents the main features of the remaining articles. 

Response

Yes

Comment

In row 217 the authors state they determined the studies appropriate for their questions, but up to this point no reference about what the actual questions of their study is made from a methodological point of view. We feel that before starting to describe the whole selection process, the authors should have a section dedicated to their study's objectives and/or research questions.

Response

The study objective is described just before the material/method section: “The objective of the study was, thus, to systematically review and synthesize empirical qualitative studies of persons with mental health conditions, healthcare providers and relatives’ subjective experiences of recovery from coercion in order to identify what helps them cope with the experience and manage its traumatic impact during and after the event; in other words, we aimed to identify the factors and processes that facilitate recovery from coercion.”

But we have added the aim/research question in section 2.1.3. (reading the studies) as a reminder: factors and processes that facilitate recovery from coercion

Comment

The whole section of findings addresses the studies and the interpretation the authors made regarding the variables they showed interest in. The whole section is very adequately referenced, and the authors provide meaningful insight about each category. This section of the study is comprehensive and extensive. 

Response

We hoped so, thank you.

Comment

The authors discuss all their findings emphasizing the categories on which recovery from coercion depends. From organizational elements to personal traits, from human relationships and interactions to the recovery process itself, the authors describe extensively all the factors identified in the scientific literature. 

Response

Yes, we tried to do so.

Comment

The section regarding the strengths and limits of this research is also well documented, the authors acknowledge elements that can be improved in future research.

Response

Thank you.

Comment

The conclusion section suggests the importance of the findings for the well-being of patients who benefit from healthcare settings. We find the results are crucial for the possibility of managing mental well-being and feel the need to congratulate the authors for their extraordinary work. 

Response

We are pleased about this comment.

Reviewer 3 Report

Comments and Suggestions for Authors

this article forms an excellent synthesis (or meta analysis) of qualitative studies on coercion measures in mental health services.

the methodology is thoroughly described and very clear to track the Author's way of thinking and managing their research.

Graphs in the article are a very good addition and make the text more understandable and coherent - this method of presenting outcomes is a major advantage of the study.

Also, the article is very well referenced and structured, and the discussion is interpretative and analytic.

I suggested Authors add a "limitations of the study section" - that was my major remark as it becomes customary to add such and it enhances methodological credibility. Aside from that I consider it a thorough and good meta analysis of the subject.

Overall the article is of high merit and importance and forms a contribution to the study of mental health care services and the use of coercion.

Author Response

Reviewer 3

Comment

This article forms an excellent synthesis (or meta analysis) of qualitative studies on coercion measures in mental health services.

Response

Thank you

Comment

the methodology is thoroughly described and very clear to track the Author's way of thinking and managing their research.

Response

Thank you

Comment

Graphs in the article are a very good addition and make the text more understandable and coherent - this method of presenting outcomes is a major advantage of the study.

Response

Thank you for commenting on this.

Comment

Also, the article is very well referenced and structured, and the discussion is interpretative and analytic.

Response

Thank you.

Comment

I suggested Authors add a "limitations of the study section" - that was my major remark as it becomes customary to add such and it enhances methodological credibility. Aside from that I consider it a thorough and good meta analysis of the subject.

Response

We have a section regarding the strengths and limits of this research, but following your remark we have noted an extra limitation: Another limitation can be that there is no specific guidance for judging the quality of a meta-ethnography; however, the transparent reporting done in this study and the feedback from the author group enhanced its trustworthiness.

Comment

Overall the article is of high merit and importance and forms a contribution to the study of mental health care services and the use of coercion.

Response

Thank you for the feedback